# PRIVACY AUDITING FOR LARGE LANGUAGE MODELS WITH NATURAL IDENTIFIERS

## ABSTRACT

The privacy auditing for large language models (LLMs) faces significant challenges. Membership inference attacks, once considered a practical privacy auditing tool, are unreliable for pretrained LLMs due to the lack of non-member data from the same distribution as the member data. Exacerbating the situation further, the dataset inference cannot be performed without such a non-member set. Finally, we lack a formal post hoc auditing of training privacy guarantees. Previous differential privacy auditing methods are impractical since they rely on inserting specially crafted canary data *during training*, making audits on already pre-trained LLMs impossible without expensive retraining. This work introduces **natural identifiers (NIDs)** as a novel solution to these challenges. NIDs are structured random strings, such as SSH keys, cryptographic hashes, and shortened URLs, which naturally occur in common LLM training datasets. Their format enables the generation of unlimited additional random strings from the same distribution, which can act as non-members or alternative canaries for audit. Leveraging this property, we show how NIDs support robust evaluation of membership inference attacks, enable dataset inference for any suspect set containing NIDs, and facilitate post hoc privacy auditing without retraining.

## 1 INTRODUCTION

Large language models (LLMs) are increasingly used in applications such as chatbots and text generation, where they are often trained on sensitive data like private conversations. This makes the need to ensure their privacy critically important. Significant research efforts have focused on both empirical and formal auditing to assess LLM privacy. Empirical audits often rely on membership inference attacks (MIAs) (Shokri et al., 2017; Carlini et al., 2022), where an adversary attempts to determine whether a particular data point was part of the model's training set. As an alternative, dataset inference Maini et al. (2021; 2024); Dziedzic et al. (2022) has emerged, which generalizes MIAs to evaluate whether an entire subset of data was used for training the LLM. On the other hand, formal auditing of claimed privacy guarantees (Steinke et al., 2023; Jagielski et al., 2020; Nasr et al., 2023), as those implemented through differential privacy (DP) (Dwork et al., 2006), attempt to empirically approximate the theoretical training guarantees.

However, recent work (Duan et al., 2024; Maini et al., 2024) has demonstrated that existing MIAs for LLMs (Mattern et al., 2023; Shi et al., 2024) are unreliable as practical tools for detecting privacy leakage. Their reported success can be largely attributed to a distribution shift between member and non-member data (Das et al., 2024), rather than their genuine ability to identify training data. When evaluated on member and non-member data drawn from the same distribution, these attacks fail to outperform random guessing (Maini et al., 2024), rendering them ineffective in realistic scenarios. This shortcoming is rooted in a broader challenge, namely the inability to obtain non-member data from the same distribution as the suspected members for most practical cases (Zhang et al., 2024a). It also affects dataset inference, which depends on access to a private validation set that matches the distribution of the suspect dataset and hinders its practical applicability.

In general, evaluating MIAs and dataset inference is challenging due to the limited availability of suitable validation data. Currently, the only available validation set comes from the Pile dataset (Gao et al., 2020), used to train the Pythia models (Biderman et al., 2023), restricting the community's ability to effectively assess their progress in detecting privacy risks or implementing protection in

more divers setups. The state-of-the-art formal post hoc privacy auditing methods (Steinke et al., 2023; Jagielski et al., 2020; Nasr et al., 2023), which aim to empirically approximate theoretical differential privacy guarantees, also cannot be applied to standard pre-trained LLMs. These methods rely on the insertion of specially crafted canary data *at training time*, a step that standard LLMs typically do not include.

As a solution to all the above-mentioned problems, in this work, we identify *natural identifiers* (NIDs). NIDs are structured random strings, generated according to some well-defined criteria, such as SSH keys, outputs from secure hash algorithms (*e.g.,* MD5 or SHA1), shortened URLs, or cryptocurrency wallet addresses. We observe that these strings are naturally included in datasets, such as code repositories (*e.g.,* GitHub) and discussion platforms (*e.g.,* StackExchange), that are used as part of the training corpora for state-of-the-art LLMs.[1] **Our unique insight is that each of the popular NIDs has a known generation function that we can leverage to generate an *unlimited* number of validation (non-member) data points from the same distribution as the NIDs which are naturally included in real-world suspect sets.**

We show how to leverage NIDs as a test-benchmark for existing and novel MIAs against pre-trained LLMs. To this end, we use the NIDs that had been included in the LLM's training data as the member set and generate further NIDs from the same type as validation set from the same distribution. These two sets can then be used to evaluate the attacks. NIDs also make dataset inference practically applicable, as one only has to identify NID types in the data subset that is suspected to be included in an LLM's training data, generate a validation set consisting of NIDs of the same type, *i.e.,* from the same distribution, and then to perform the dataset inference procedure. We empirically analyze this approach in a controlled environment, using open-source LLMs and their known training data. Specifically, we use the Pythia suite of models with the Pile dataset and the OLMo models. For OLMo models, we extend their training data, the Dolma dataset, with a post-hoc validation set using the identified NIDs. Across all the data subsets, our NID-based dataset inference successfully achieves p-values below $0.1$ for distinguishing between training and validation data splits. Additionally, it does not falsely identify data as being used during training, *i.e.,* we do not observe any false positives, with p-values exceeding $0.5$, when the validation set from the Pile is selected as the suspected set.

The NIDs also enable us to perform post hoc privacy auditing for LLMs. We build on the currently fastest single training run auditing approach (Steinke et al., 2023), which needs to include dedicated canaries in the training set. We demonstrate that when NIDs naturally occur in the training set, we can construct the auditing set of NIDs from the same type post hoc and retroactively assess the privacy guarantees of any LLM without the requirement of retraining from scratch. This alleviates the prohibitively expensive retraining and makes auditing practical for existing models.

In summary, we introduce NIDs as the solution to three pressing challenges in LLM privacy research and practical privacy assessment. Utilizing NIDs, we construct a test bench with member and non-member data from the same distribution to systematically evaluate the performance of existing and future MIAs and dataset inference approaches on diverse state-of-the-art LLMs. We demonstrate how to leverage the NIDs to perform dataset inference in practical scenarios and to conduct truly post hoc privacy auditing. Through extensive empirical evaluation of the Pythia suite and the Pile dataset, we demonstrate the effectiveness of NIDs as a tool for auditing and analyzing privacy risks in LLMs.

## 2 BACKGROUND

**Membership Inference** (MI) (Shokri et al., 2017) aims to determine whether a specific data point was included in a model's training set. MI has diverse applications, and in this work, we focus on their use for privacy auditing (Steinke et al., 2023). While Membership Inference Attacks (MIAs) have been extensively explored for small scale models, MI for LLMs is a much more challenging problem. The latest work Duan et al. (2024); Maini et al. (2024); Zhang et al. (2024a) indicates that the success reported by previous MIAs on LLMs (Mattern et al., 2023; Shi et al., 2024) is rather due to a distribution shift than to the attacks' ability to distinguish between the member and non-member

---

[1]Indeed, we observe that the publicly available datasets used to train popular LLMs, such as the Pile (Gao et al., 2020) or Dolma (Soldaini et al., 2024), contain 30637 and 23571 different types of NIDs, respectively— showcasing the practical availability of NIDs. This large number of NID-types and new types constantly emerging, makes it impossible to omit them through the web crawlers, thus NIDs are less prone to be excluded from the LLMs' training set.

data. A prominent example is temporal distribution shifts that occur when data before a specific cutoff date is selected as members and data after the point is treated as non-members and both differ in language, wording, or formatting styles. When evaluated in the correct setting without distribution shift, Maini et al. (2024) showed that most attacks do not outperform random guessing. Another issue when using MIAs for auditing LLM training is their need for shadow models (Meeus et al., 2024; Eichler et al., 2024; Carlini et al., 2022), *i.e.,* models with the same architecture trained on different splits of the data, which become prohibitively expensive to train as LLMs grow in size. To address these challenges, we create a new benchmark based on NIDs to provide a robust evaluation of MIAs on LLMs.

**Dataset Inference** (DI) (Maini et al., 2021) aims to resolve whether a given dataset was used to train a model. Thus, in comparison to MIAs, DI operates on the dataset level and was initially designed to protect the model's ownership. The core idea in the original method, designed for supervised learning, is that classifiers tend to repel *training examples* further from decision boundaries, whereas *test examples*, having no impact on the model's parameters, remain closer to these boundaries. This concept was later adapted to self-supervised learning (SSL) models (Dziedzic et al., 2022), leveraging the insight that the representations of training data induce substantially different distribution then representations of test data. Dataset inference was also extended to LLMs (Maini et al., 2024), enabling the detection of datasets used during their training.

However, DI always relies on an access to a *private validation set from the same distribution as a suspect set*. Prior work (Zhang et al., 2024a) argues that this makes DI inapplicable for real-world use-cases where such data is usually not available. As a solution, they propose to inject random and meaningless canaries into the data and then test how the LLM ranks the selected canary among all alternatives. Since they assume access to the generator of the random canaries, they can provide the corresponding validation data points and avoid distribution shifts. The approach's reliance on fully random strings might also reduce the practical applicability of this approach since content creators would have to artificially include such specialized strings into their datasets and hide them from human readers. Additionally, web crawlers can be trained to omit such arbitrary context-free strings when scraping the data from the Internet, reducing the likelihood of this data to getting included into LLMs' training data. Finally, this solution does not work for already existing LLMs which were trained without injected canaries. In contrast, our observation is that we can leverage NIDs that naturally are included in LLMs' training sets, mitigating the necessity from inserting purely random strings and enabling auditing of existing pre-trained LLMs without retraining.

**Differential Privacy** (DP) (Dwork et al., 2006) is a mathematical framework that provides a rigorous framework for limiting privacy leakage, ensuring that no individual's data significantly impacts the outcome of a computation. Formally, a randomized mechanism $M$ satisfies $(\varepsilon, \delta)$-DP if, for any two inputs $x$ and $x'$ differing by a single individual's data and any measurable set $S$, the following holds:

$$P[M(x) \in S] \le e^\varepsilon P[M(x') \in S] + \delta.$$

In this definition, $\varepsilon$ bounds the privacy leakage, while $\delta$ represents the probability of this bound to fail.

**Auditing DP.** Privacy audits attempt to empirically estimate a lower bound on the privacy parameters $\varepsilon$ and $\delta$ post training. These audits help evaluate the tightness of the theoretical analysis (Jagielski et al., 2020; Nasr et al., 2023) and can also reveal errors in the mathematical analysis or flaws in the algorithm's implementation (Tramer et al., 2022). In general, privacy auditing relies on retraining models and inserting canaries during training (Jagielski et al., 2020; Nasr et al., 2023; Steinke et al., 2023). While Steinke et al. (2023) limit the computational overhead by proposing a privacy auditing technique that can operate with a single training run, for large LLMs with trillions of parameters, even this might be prohibitively expensive. We build on their approach and leverage NIDs to remove the need for retraining altogether.

**Canary Exposure** (Carlini et al., 2019) is a method to rigorously quantify data leakage in machine learning models. It relies on inserting random sequences called *canaries* into the model's training dataset. Then, it measures *exposure* of this data point as the decrease in perplexity of the model on this data vs other similar random sequences not seen during training. Formally, exposure can be defined as follows: Given $m$ *canaries* $C = \{c\}_i^m$, which are sampled and added into the model's training set, and $n$ *references*, *i.e.,* other random strings, $R = \{r\}_i^n$ that are sampled and withheld for comparison, we compute the exposure for each canary $c_i$ using the rank of its loss, $\ell(c_i)$, among the

losses of all references $r_i$, in the following way:

$$\text{Exposure}(c_i) = \log_2(n) - \log_2(\text{Rank}(\ell(c_i), \{\ell(r_i)\}_{i=1}^n)).$$

The rank assigns a value ranging from 1 (indicating that $c_i$'s loss is lower than all references) to $n + 1$ (indicating that $c_i$'s loss is higher than all references). The higher the exposure is, the more likely that the model was trained on the given canary. The rank of 1 and corresponding exposure of $log_2(n)$ indicates the highest memorization of the data in the model. In the case of no memorization, assuming high values of $n$, the rank will be around $n/2$, and the corresponding exposure of $ln$. We leverage this *canary exposure metric* for our privacy auditing since it can be interpreted with respect to the true positive ratios (TPRs) and false positive ratios (FPRs) of MIAs (Jagielski, 2023), enabling estimation of a lower bound on privacy leakage.

## 3 NATURAL IDENTIFIERS (NIDS)

We introduce NIDs, explore their natural occurrence, and provide the intuition on how they address key challenges in LLM privacy research. We then present the notation and formalization of NIDs, which will serve as the foundation for the subsequent sections.

### 3.1 NIDS IN THE WILD

On a conceptual basis, NIDs are structured random strings, generated according to some well-defined function. Prominent examples include SSH keys, outputs from secure hash algorithms (*e.g.,* MD5 or SHA1, SHA256), shortened URLs, or cryptocurrency wallet addresses. Such strings are omnipresent on the internet, *e.g.,* in code repositories (*e.g.,* GitHub) and discussion platforms (*e.g.,* StackExchange).

Since large parts of the data used to pre-train state-of-the-art LLMs are simply crawled from the internet, these NIDs get naturally included in the LLMs' training sets. We analyzed a wide range of popular LLM training datasets, including the Pile (Gao et al., 2020) and Dolma (Soldaini et al., 2024), and identified that all of them contain multiple types of NIDs with many examples per type. We provide an overview on the analyzed subsets and contained NIDs in Table 2.

The main reason why these (partially random) strings are not removed from the data by the web crawler when composing the dataset is the severe difficulty of identifying them. This results from the fact that in contrast to truly random strings, such as the canaries by Zhang et al. (2024a), NIDs can carry a meaning in their given context. Additionally, new types of NIDs, *e.g.,* produced through novel URL shortening approaches, are emerging continuously. Hence, even when using regex filtering on currently known NIDs, a significant amount of (potentially new) NIDs are likely to remain in the datasets. In fact, we observe in Table 2 that even highly filtered and curated datasets, such as Dolma (Soldaini et al., 2024) contain significant fractions of NIDs. This makes our solutions for LLM privacy based on them stealthy and widely applicable.

### 3.2 LEVERAGING NIDS

What makes NIDs special is their rigorously specified format in combination with a sequence of random characters. Given that their format is known, it becomes possible to generate an *infinite number* of other random strings that follow the same distribution. In the following, we present the intuition on how this property contributes to solving three of the most pressing challenges in LLM privacy research. Further details and formalization of the respective problems and the solution enabled through NIDs are presented in the next sections, respectively.

**1) NIDs provide a MIA-benchmark.** As discussed in the previous section, the progress by current MIAs is hard to measure because of the lack of non-member data from the exact same distribution as the member data. Additionally, due to the LLMs' sheer sizes, retraining the models or shadow copies for MIAs becomes prohibitively expensive, limiting evaluation further. NIDs can overcome both limitations and be used to provide a benchmark for existing and future MIA attacks. Given that state-of-the-art pretrained LLMs have NIDs in their training data, we can *generate* a large set of validation data from exactly the same distribution. Using this validation set and its corresponding NID-based training members, novel and existing MIAs can be evaluated *without distribution shift*

Table 1: **NID-benchmark for Pythia-12b.** The AUC for MIAs between the NIDs and the corresponding GIDs on various subsets of the Pile dataset.

| MIA | Pile | | Github | | StackExchange | | UbuntuIRC | Wikipediaen | Train | | | | Average | |
|---|---|---|---|---|---|---|---|---|---|---|---|---|---|---|
| | Train | Test | Train | Test | Train | Test | | | PubMedCentral | HackerNews | Pile-CC | ArXiv | Train | Test |
| Loss | 58.6 | 50.3 | **71.8** | 51.1 | 50.3 | 50.9 | 50.3 | 50.6 | 50.6 | 60.5 | 51.1 | 50.4 | 54.9 | 50.7 |
| Min-K% | 57.6 | 51.0 | 68.4 | 50.6 | 50.7 | 51.2 | **51.1** | 50.6 | 50.7 | 60.5 | 52.3 | **51.0** | 54.8 | 50.9 |
| Min-K%++ | 56.9 | **51.4** | 71.2 | 50.3 | **50.8** | **51.9** | 51.1 | 51.3 | **51.1** | **69.7** | **53.2** | 50.9 | **56.2** | **51.2** |
| ReCALL | 53.5 | 50.2 | 50.6 | 50.3 | 50.0 | 51.1 | 50.3 | 51.3 | 50.2 | 57.8 | 50.1 | 50.2 | 51.6 | 50.5 |
| ReCALL (Hinge) | 51.3 | 50.1 | 53.3 | 50.4 | 50.4 | 51.4 | 50.5 | **51.9** | 50.8 | 50.3 | 50.4 | 50.0 | 51.0 | 50.6 |
| Hinge | **58.7** | 50.5 | 71.8 | **51.5** | 50.4 | 50.5 | 50.4 | 50.4 | 50.5 | 60.8 | 50.9 | 50.4 | 54.9 | 50.8 |

and *without retraining*. The biggest advantage is that this approach allows to eventually assess MIAs on a wide range of state-of-the-art existing pre-trained LLMs, namely all of them that hold some NIDs in their training data—providing a broad attack evaluation setup. We showcase the usefulness of NIDs as a MIA-benchmark in Section 4.

**2) NIDs enable DI.** With the same reasoning, NIDs enable DI for any suspect set (*i.e.,* a dataset for which we want to assess whether it has been used to train a given LLM (Maini et al., 2024)) that contains NIDs. Again, we can generate a set of IDs that follow the exact same distribution as the NIDs in the suspect set and use them as a validation set for the DI. In case the LLM has been trained on the suspect set, it will react differently on the NIDs included in the suspect set and their generated counterparts from the validation set. Otherwise, it it was not trained on the suspect set, its behavior will be the same over both sets, as both NIDs and their generated counterparts will just be the same type of random strings for the LLM. Thereby, it is possible to identify whether the suspect set was indeed used to train the model. We detail the use of NIDs for DI in Section 5.

**3) NIDs facilitate post hoc privacy audits.** Finally, we can use NIDs to perform a post hoc privacy audit for LLMs trained with DP, as long as there are NIDs in the LLMs' training data. To do so, we build on the one-run privacy audit by Steinke et al. (2023). In their method, they select a set of data points to be included or excluded during a training run. After training, an auditor attempts to infer whether each data point was included or excluded, with the option to abstain from guessing in uncertain cases. The fraction of correct guesses provides a lower bound on the privacy parameters. Using our NIDs, it is no longer necessary to retrain the model. Instead, we generate random samples from the same distribution as the ones seen during training. The NIDs as natural canaries can be ranked against the generated ones, with respect to their exposure, for auditing *without any retraining*, *i.e.,* truly post hoc.

## 3.3 Formalizing NIDs

An *identifier (ID)* $v$ is constructed in the following way $v := W(z), z \in \mathcal{Z}$, where $z$ is a random sequence that comes from a known independent distribution $\mathcal{Z}$ (or more generally a source of randomness), and $W$ is a generation function. Additionally, we define a set of IDs, generated by a generation function $W$, as $V := \{W(z) : z \in \mathcal{Z}\}$. A *Natural Identifier (NIDs)* is an ID that is part of a real dataset. Given an NID, using the corresponding generation function $W$, we can generate many new IDs, which we refer to as *Generated Identifiers (GIDs)*. GIDs are IDs that are not part of any real dataset, but generated based on an NID.

As a concrete example, to generate the RSA (Rivest et al., 1978) private and public keys, we provide a pair of two randomly selected prime numbers $p$ and $q$, thus, in this case, our $z = pq$. Then, given a NID, which represent a public RSA key, can use the corresponding generation function $W(z) := \text{RSA}(z)$ to generate new GIDs. In this case, the set $V$ is the set of all the public RSA keys. The main property of NIDs is that a priori each ID $v \in V$ is equally likely to be generated and published because it only depends on the source of randomness and not on the context. Note that the generation function $W$ might and will likely depend on the context. The second important property of NIDs is that they allow easy sampling from the set $V$. In the suspect datasets $D_{\text{sus}}$, which we are auditing, there are usually $m$ NIDs, with the corresponding sets $V_1, \ldots, V_m$. Furthermore, for each set $V_i$ where $i \in [m]$, we denote the NID as $\hat{v}_i \in V_i$, and specifically, the NID that belongs to the suspect dataset as $\hat{v}_i \in D_{\text{sus}}$. Finally, we define $\Sigma_i$ as the set of all the permutations over $V_i$.

## 4 NIDs for Benchmarking MIAs

In this section, we analyze NIDs' potential to serve as a MIA-benchmark. As examples for models of various sizes and families, we experiment with Pythia-2.8b, Pythia-6.9b, Pythia-12b, and OLMo-7B. The Pythia models are pre-trained on the Pile dataset, which consists of various subsets. OLMo-7B is trained on the Dolma dataset.

We analyze the subsets for the occurrence of NIDs (see Table 2 in Appendix B) and identify that the subsets with code, such as Stack Exchange and GitHub, and large non-topic-specific corpus, such as Refined Web and Pile Common Crawl, have a high number of NIDs. SHA1 and MD5 are overall the most frequent types of NIDs For some large subsets, such as Refined Web, we have as many as 16989 NIDs, however, for smaller subsets, the number is smaller. For instance, the whole validation and test set of the Pile is around 2 GiB, and we detected 293 NIDs, 197 of which are in the GitHub subset.

In our NID benchmark, for each NID, we generate 127 new GIDs. We choose 127 new samples to strike a good balance between the computational cost of evaluation of too many samples and a good estimate of the generated sample distribution. By construction, these newly GIDs are non-members (from the same distribution as the member NIDs), and can, thereby, be used to evaluate the success of MIAs. Strong MIAs should have a high performance, *e.g.,* measured in AUC score, when presented with NIDs from the LLM's training set and their generated counterparts. In contrast, for NIDs not present in the LLM's training set and their generated counterparts, the success should be similar to random guessing, *i.e.,* an AUC score of around $0.5$.

Using our identified NID member set and the respectively generated non-member set, we evaluate existing MIAs for LLMs, namely Loss (Yeom et al., 2018), Min-K% (Shi et al., 2024), Min-K%++ (Zhang et al., 2024b), ReCall (Xie et al., 2024), and Hinge (Carlini et al., 2022). We present the results on Pythia-12b in Table 1. While, in Appendix D, we show the results for OLMo-7B[2], Pythia-6.9b and Pythia-2.8b in Table 3, Table 4 and Table 5, respectively. Additionally, in Appendix D, we report the performance of each model using TPR@1% FPR.

We verify the performance on the train and test sets. For most MIAs, we observe that the performance is extremely close to random guessing on the test set, following the expected behavior and indicating that there is, indeed, no distribution shift between the NID members and our generated non-members. Additionally, the results are well-behaved in the sense that the average AUC on the train set is noticably higher than on the test set. Contrary to what was suggested by the Xie et al. (2024), the ReCALL attack does not provide the best performance, while the improvement shown by Min-K%++ (Zhang et al., 2024b) translates to our settings.

The evaluation on our benchmark also validates the findings by prior work (Maini et al., 2024; Das et al., 2024), in particular the ones made based on the MIMIR (Duan et al., 2024) dataset—a dataset, derived from the Pile train and validation sets proposed as an evaluation dataset for MIAs: Namely, without a distribution shift between the member and non-member data, most existing MIAs for LLMs do not perform much better than random guessing for most of the datasets. Yet, conceptually, compared to the MIMIR dataset, using NIDs allows us to evaluate MIAs and LLMs without requiring a validation set by leveraging generated IDs to create new samples that closely resemble the original ones. This key difference enables us to evaluate a worst-case privacy scenario, which is more rigorous than relying on a random train-validation splits, which is an essential factor when auditing privacy leakage (Aerni et al., 2024).

Notably, we are the first to assess MI performance on OLMo-7B. We observe that OLMo-7B, which has been trained on 2.05T tokens, is much more robust to MIAs compared to the Pythia models (*e.g.,* 6.9B), which have been trained on 300B tokens. This result validates the trend that models trained on larger corpus memorize less (Maini et al., 2021).

Overall, our results highlight that NIDs enable benchmarking MIAs and evaluate the privacy leakage of LLMs without the problems of distribution shift, even without any additional calibration techniques required by prior work (Carlini et al., 2022; Watson et al.), and without the computational costs of retraining. Thereby, our NID-benchmark enables to practically evaluate MIAs on various large state-of-the-art LLMs without the need of a validation set.

---

[2]https://huggingface.co/allenai/OLMo-7B-0424-hf

## 5  DATASET INFERENCE WITH NIDS

Next, we turn to exploring the use of our generated same-distribution data for performing DI (Maini et al., 2021). As discussed in Section 2, the strongest limitation of DI is its reliance on a private validation set from the same distribution as the suspect dataset, *i.e.,* the dataset for which we want to assess whether it was included in the training of the given model. Such datasets are often not available in practical applications (Zhang et al., 2024a). We present how our NIDs can overcome this limitation and enable successful DI for suspect datasets that contain NIDs.

When given a suspect set $D_{\text{sus}}$, we first need to identify and extract which NIDs are included. Please refer to Appendix A for more details on this process. The extracted NIDs form the suspect subset $D'_{\text{sus}}$ which we use to perform the DI. Then, for every real NID in $D'_{\text{sus}}$, we generate 127 new GIDs with the same NID type and with the same structure to form the validation set from the same distribution as $D'_{\text{sus}}$.

In terms of executing DI, we closely follow Maini et al. (2024). To extract features from the suspect set $D'_{\text{sus}}$ and our validation set, we run the state-of-the-art MIAs. We extract features based on Loss (Yeom et al., 2018), Min-K% (Shi et al., 2024), Min-K%++ (Zhang et al., 2024b), and ReCall (Xie et al., 2024). Next, following the DI protocol, we need to learn the correlation between the features (the MIA scores), and their membership status. To learn this correlation, we train a gradient boosting trees classifier to distinguish between the two distributions. To use all the samples available, we train and score the samples using K-Fold, and we ensure that the generated samples derived from a real sample end up in the same fold. Finally, following Maini et al. (2024), we perform statistical testing and compute the p-values. Under the null hypothesis, which assumes that NIDs are not part of the training data, the ranks of each NID relative to its corresponding GIDs should follow a uniform distribution. This means that if we order the NIDs based on their association with GIDs, their positions should be evenly distributed across the ranking scale. To test this assumption, we apply the Kolmogorov–Smirnov (KS) test. If the KS test detects a significant deviation from uniformity, we reject the null hypothesis, suggesting that the NIDs may, in fact, be present in the training data. Small p-values indicate that we can reject the null hypothesis, *i.e.,* we are confident that the model was trained on the suspect set. Large p-values suggest that the test was inconclusive and we are not confident whether the model was trained on the suspect set or not.

Using our generated validation set with GIDs and the suspect set $D'_{\text{sus}}$ with NIDs, we perform DI on various models and data subsets. We take relatively small suspect sets $D'_{\text{sus}}$ with 100 real NIDs to simulate a realistic setup, and we only consider subsets with at least 100 NIDs, with the only exception of Proof Pile 2 Test, which has only 85 samples, however, it is the only test set available for Dolma. For each subset, we generate a validation set using the NIDs, and perform DI. We consider 28 training and 7 test subsets across 4 models (Pythia-2.8b, Pythia-12b, Pythia-6.9b, and OLMo-7B). Our method shows that for the suspect sets that were included in the training data, DI obtains low p-values ($< 0.01$) that allow to reject the null hypothesis. This highlights that the suspects are correctly identified as training data. At the same time, for test data, *i.e.,* datasets that were not used to train the given LLM, we observe high p-values that do not allow us to reject the null hypothesis. The sets are, hence, correctly not marked as training data (p-values $>> 0.01$).Table 10 in Appendix E shows the p-values for each dataset and model.

We present further results on models of various sizes and with varying numbers of NIDs in the suspect set in Figure 3 of Appendix C. The results highlight that the more NIDs are available in $D'_{\text{sus}}$, the more reliable the DI. Overall, using NIDs and the generated validation set, we observe no false positives, while correctly identifying training subsets (true positives). This highlights NIDs' ability to enable DI on suspect datasets that contain NIDs.

## 6  DP AUDITING WITH NATURAL IDENTIFIERS

Using our NIDs, we adapt the method proposed by Steinke et al. (2023) to create a novel post-hoc DP auditing. Their technique considers $m$ canary samples and uses coin flips to randomly determine which samples should be included in the training set. Therefore, it is a binary case of adding or removing a single sample (and selecting between two options). In our framework for extending their method to post hoc audits using NIDs, we first identify the NIDs that were present in the training data and denote their total number as $m$. For each NID $i \in [m]$, we generate the corresponding GIDs,

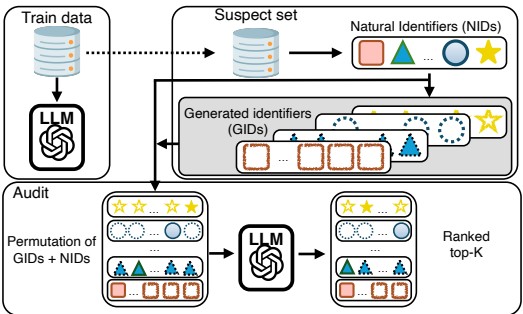

Figure 1: **Post-hoc DP auditing with Natural Identifiers.** (1) A third party trains the model using the training data. (2) Given a suspect dataset, we filter the NIDs from the real training dataset. (3) We generate the new GIDs using the NIDs. (4) We join NIDs with corresponding GIDs and permute each set. (5) We rank the samples based on the model's outputs choosing top-k.

and the corresponding set of IDs $V_i$. One of the main properties of NIDs is that, a priori, any element in $V_i$ could have been part of the training data in place of the NID. This enables us to model privacy auditing analogously to the fixed-length dataset variant proposed by Steinke et al. (2023). The key distinction in our approach is that, rather than selecting between two alternatives prior to training, we consider the NIDs as inserted canaries with the GIDs as multiple left-out canary possibilities for each set $V_i$. Figure 1 summarizes how to leverage the NIDs to audit DP post hoc. We consider the NIDs as the input to a training procedure $M$ (also referred to as the mechanism), which may satisfy $\varepsilon$-DP. Given the resulting trained model, an auditor seeks to infer, for each set $V_i$, which sample was the NID and was included in the training data. To do so, the auditor ranks the samples in $V_i$ from the most to the least likely candidate to be the NID. A prediction is considered correct if the true NID appears among the top-$r_i$ ranked samples, where $r_i$ is a predefined threshold.

Following the analysis of Theorem 5.2 by Steinke et al. (2023), we can adapt their privacy auditing procedure to our setting. Similarly to the standard exposure setting, we compare the rank of the real samples and alternative samples.

**Theorem 1.** *Let $M : V_1 \times ... \times V_m \to \Sigma_1 \times ... \times \Sigma_m$ be an $\varepsilon$-DP mechanism under replacement. Let $S \in V_1 \times ... \times V_m$ be uniformly random, and define $T = M(S) \in \Sigma_1 \times ... \times \Sigma_m$. Then, for all $v \in \mathbb{R}$, all $t \in \Sigma_1 \times ... \times \Sigma_m$ in the support of $T$, and all $r_1, ..., r_m$ with $r_i \leq |V_i|$,*

$$\mathbf{P}_{\substack{S \leftarrow V_1 \times ... \times V_m, \\ T=M(S)}} [\sum_{i=1}^{m} \mathbb{1}[\mathrm{rank}(t_i, S_i) \leq r_i] \geq v | T = t]$$

$$\leq \mathbf{P}_{\hat{S} \leftarrow \mathrm{Bernoulli}(\frac{r_i e^\varepsilon}{|V_i|-1+e^\varepsilon})_{i=1}^{m}} [\hat{S} \geq v] := \beta(k, \varepsilon, v, t, r)$$

$\mathrm{rank}(a, b)$ *returns the 1-based position of $b$ in the permutation $a$, where $a$ is permutation and $b$ is an element.*

In our setting, Theorem 1 states that if the LLM is trained with $\varepsilon$-DP, any attacker attempting to detect the NID is constrained. Concretely, the attacker ranks the LLM's output on both the NID and its corresponding GIDs from most to least likely. Then, they count how many NIDs appear in the top-$r$ where $r$ s a predefined threshold. The theorem states that this count is bounded by a Bernoulli distribution, whose probability depends on $\varepsilon$, $r$, and the number of GIDs. This theorem enables DP auditing through its hypothesis-testing interpretation: under the null hypothesis that the LLM is $\varepsilon$-DP, we can derive a confidence interval for the lower bound on $\varepsilon$. We present the full proof of Theorem 1 in Appendix F. To demonstrate the effectiveness of our auditing, we apply it to the randomized response mechanism (see Appendix G).

While Theorem 1 is specific to $\varepsilon$-DP, most of the existing private deep learning algorithms, such as DP-SGD (Abadi et al., 2016), focus mostly on $(\varepsilon, \delta)$-DP. Therefore, following the analysis by Steinke et al. (2023), we also adapt the $(\varepsilon, \delta)$-DP auditing to our setting (see Theorem 2 in Appendix F).

**Evaluating our Privacy Auditing.** We verify that our proposed framework applies to privacy auditing in LLMs by adapting the black-box procedures from Steinke et al. (2023) in the fixed-size

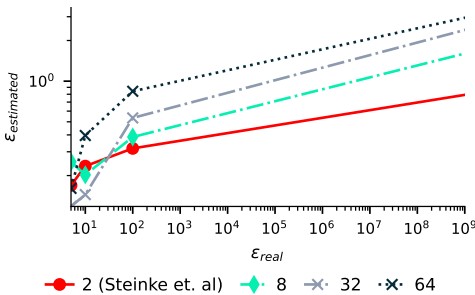

Figure 2: **Impact of cardinality** ($c = \{2, 8, 32, 64\}$) **on $\varepsilon$ estimation.** Experiments were conducted using $\varepsilon$ values of $\{5, 10, 100, \infty\}$.

dataset variant. The auditing process follows the algorithm described in Appendix H.2. To fully control the auditing process, we create canaries using PersonaChat (Jandaghi et al., 2023), an AI-generated dataset containing dialogues of people describing themselves. Specifically, we crafted $m$ NIDs by adding either SHA512 or SHA256-like sequences at the beginning of real text sequences to construct the training set. By crafting our NIDs this way, we want to control their insertion, frequency, and uniqueness precisely, thus ensuring the complete coverage of our assumption while still emulating a realistic setting. Then, for each NID, we generate $c$ GIDs. In this way, we have sets of IDs $V_1, \ldots, V_m$. We train Pythia 70M (Biderman et al., 2023) with full fine-tuning using DP-SGD (Abadi et al., 2016) with $\delta = 10^{-5}$ for 20 epochs using the maximum sequence length of 64 tokens. As a $SCORE$ function (see Algorithm 1), we use Min-K% (Shi et al., 2024) and Loss to determine the best estimated $\varepsilon$ value. For all of the experiments, we use $r_i = 1$ for ranking, meaning that the guess is correct only if the most likely prediction given by the attacker is the real NID.

As a reference, we use the fixed-length dataset auditing introduced by Steinke et al. (2023), a special case of our method, where $|V_i| = 2$ and $r_i = 1$. The empirical analysis in Figure 2 demonstrates that our method outperforms the baseline across multiple cardinality parameters ($c \in \{8, 32, 64\}$) in fixed-length dataset settings. While higher cardinality can enhance the statistical power of the auditing procedure in the best-case scenario—meaning fewer samples are required—the ranking task becomes increasingly complex. Instead of merely comparing two candidates, one must select from $c$ options. For smaller privacy budgets (*i.e.,* a more challenging prediction task), smaller cardinality is beneficial, whereas for larger $\varepsilon$, higher cardinality tends to be advantageous, thus significantly over-performed the baseline as $\varepsilon$ increases. This trend follows our considerations for randomized response (see Appendix G), where increasing cardinality improves utility, particularly in less restrictive privacy settings. For a deeper discussion on the impact of the number of inserted canaries and the choice of SCORE function on auditing tightness, see Appendix H.1.

Finally, we note that our auditing approach provides a lower bound on privacy leakage, focusing on NIDs. While it may not capture the worst-case memorization, it offers a tighter bound than the original method in realistic scenarios.

## 7    DISCUSSION AND CONCLUSIONS

We introduce the concept of *natural identifiers* (NIDs) and demonstrate how they address three pressing challenges in LLM privacy research: (1) the difficulty of evaluating LLM MIAs without introducing distribution shifts between members and non-members, (2) the inapplicability of DI when no validation dataset from the same distribution as the suspect set is available, and (3) the limitation in privacy audits due to existing methods' reliance on retraining. Although we focus on leveraging NIDs within the language domain for models trained on datasets containing NIDs, our analysis highlights that most standard LLM pretraining datasets naturally include a diverse and extensive set of NIDs. This broad presence makes NIDs widely applicable. Our thorough empirical evaluations with multiple state-of-the-art LLMs underline this insight and show the practical benefits of leveraging NIDs to benchmark MIAs, enable DI in real-world scenarios, and perform truly post hoc privacy audits without retraining. We believe these contributions will significantly advance LLM privacy research by enabling computationally efficient and effective privacy evaluations.

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

## A   EXTRACTING *natural identifiers* POST-HOC

In this section, we describe how to extract *natural identifiers* robustly. First, we select a series of regular expressions to identify potential *natural identifiers*. Depending on the type of secret, there might be a high number of false positives, therefore, we need to further remove invalid samples. We achieve that by first removing duplicates and then running a blind baseline (Das et al., 2024; Zhang et al., 2024a) using the n-grams as features and different types of tabular classifiers, such as Naive Bayes classifier, Gradient Boosting Trees and Logistic Regression. Via K-Fold, we compute the MI score of each sample, then, we compare the rank of the real sample with respect to the generated ones. If the rank of the generated sample is too low or high, we discard that sample.

We follow this procedure to filter invalid *natural identifiers* robustly. For instance, strings with "012456789" are unlikely to be random strings and are mostly likely false positives. Finally, we check that the final blind baseline performance is close to random guessing, and the sample is particularly predictive using a blind baseline.

For each type of NID, we have a specific way to generate them to closely resemble the original sample. **MD5.** We generate the samples uniformly using this condition `[a-fA-F0-9]{32}` following the sample casing.
**SHA1.** We generate the samples uniformly using this condition `[a-fA-F0-9]{40}` following the same casing of the original sample.
**SHA256.** We generate the samples uniformly using this condition `[a-fA-F0-9]{64}` following the same casing of the original sample.
**SHA512.** We generate the samples uniformly using this condition `[a-fA-F0-9]{128}` following the same casing of the original sample.
**Ethereum wallet.** We generate the samples uniformly using this condition `0x[a-fA-F0-9]{40}`. We select and generate only samples using case sensitivity as a checksum (ERC-55: Mixed-case checksum address encoding).
**Java serialization.** All serializable Java classes have the `serialVersionUID` attribute, which is often equal to a random number, for instance, `private static final long serialVersionUID = 6146619729108124872L`.

## B   DISTRIBUTION OF NATURAL IDENTIFIERS

Table 10 shows for each subset and type of NID the number of NIDs. We highlight that large subsets, such as Dolma RefineWeb, has significant number of NIDs.

## C   FURTHER EXPERIMENTS ON DI

We evaluate DI on various models and data subsets. More concretely, we experiment with Pythia models 12b, 6.9b, and 2.8b and OLMo-7B. Additionally, we investigate the impact of increasing the number of samples in the suspect set. All results are summarized in Figure 3.

## D   MIAS PERFORMANCE

Table 1, Table 4 and Table 5 show the MI performance of the individual MIAs on the subsets of the Pile using the NIDs, where the goal is to distinguish the real from the generated ones. Furthermore, for completeness, we have Table 6, Table 7, Table 8, that show the MI performance using TPR @ 1% FPR.

## E   DI P-VALUES

Table 10 shows the p-values of the DI task in the different models and datasets. Our method shows no false positives and no fal

Table 2: **Natural Identifiers in Different Datasets.** We present the number of various *natural identifiers* (here: sha1, md5, sha256, java serialization, sha512, and ethereum wallet) in the analyzed datasets. The *sum* denotes the total number of *natural identifiers* in a given dataset.

| Dataset | Total Number | sha1 | md5 | sha256 | java serialization | sha512 | ethereum wallet |
|---|---|---|---|---|---|---|---|
| dolma refineweb | 16989 | 8098 | 6192 | 2130 | 42 | 110 | 417 |
| pile train github | 13182 | 5389 | 1938 | 4158 | 819 | 701 | 177 |
| pile train stackexchange | 9862 | 4850 | 3235 | 1200 | 348 | 121 | 108 |
| pile train pile cc | 3422 | 1078 | 2008 | 274 | 1 | 8 | 53 |
| dolma algebraic stack train | 2384 | 1264 | 464 | 612 | 1 | 28 | 15 |
| pile train hackernews | 2268 | 1340 | 821 | 93 | 0 | 7 | 7 |
| dolma open web math train | 2207 | 1212 | 727 | 221 | 1 | 20 | 26 |
| pile train ubuntuirc | 1056 | 618 | 340 | 88 | 0 | 9 | 1 |
| dolma c4 | 791 | 408 | 301 | 63 | 0 | 4 | 15 |
| dolma PeS2o | 435 | 235 | 174 | 11 | 0 | 1 | 14 |
| dolma MegaWika | 383 | 115 | 200 | 62 | 0 | 2 | 4 |
| dolma ArXiv | 332 | 239 | 58 | 21 | 0 | 2 | 12 |
| Pile test (all subsets) | 293 | 130 | 69 | 62 | 13 | 14 | 5 |
| pile train pubmedcentral | 225 | 66 | 152 | 7 | 0 | 0 | 0 |
| pile train ArXiv | 207 | 75 | 122 | 7 | 0 | 0 | 3 |
| pile test github | 197 | 80 | 36 | 52 | 13 | 12 | 4 |
| pile train wikipediaen | 85 | 15 | 66 | 3 | 0 | 1 | 0 |
| pile test stackexchange | 58 | 34 | 16 | 6 | 0 | 2 | 0 |
| open web math test | 46 | 19 | 20 | 6 | 0 | 1 | 0 |
| algebraic stack test | 39 | 28 | 4 | 7 | 0 | 0 | 0 |
| dolma wiki | 38 | 11 | 22 | 3 | 0 | 2 | 0 |
| pile test pile cc | 18 | 6 | 8 | 3 | 0 | 0 | 1 |
| pile train philpapers | 16 | 1 | 15 | 0 | 0 | 0 | 0 |
| pile train freelaw | 15 | 1 | 14 | 0 | 0 | 0 | 0 |
| pile test hackernews | 13 | 7 | 6 | 0 | 0 | 0 | 0 |
| dolma tulu flan | 10 | 0 | 9 | 1 | 0 | 0 | 0 |
| pile test ubuntuirc | 5 | 3 | 2 | 0 | 0 | 0 | 0 |
| pile train enronemails | 4 | 0 | 4 | 0 | 0 | 0 | 0 |
| pile test wikipediaen | 2 | 0 | 1 | 1 | 0 | 0 | 0 |
| dolma books | 2 | 0 | 2 | 0 | 0 | 0 | 0 |
| pile train gutenbergpg 19 | 1 | 0 | 1 | 0 | 0 | 0 | 0 |
| pile train pubmedabstracts | 1 | 0 | 1 | 0 | 0 | 0 | 0 |

Table 3: **NID-benchmark for OLMo-7B.** The AUC for MIAs between the NIDs and the corresponding GIDs on various subsets of the Dolma dataset.

| MIA | C4 | PeS2o | MegaWika | ArXiv | refineweb | algebraic stack | open web math | Proof Pile 2 Test | Average Train |
|---|---|---|---|---|---|---|---|---|---|
| | | | | | Dolma | | | | Average Train |
| Loss | 50.1 | 50.2 | 50.2 | 51.2 | 50.1 | 50.0 | 50.9 | 50.6 | 50.4 |
| Min-K% | 50.1 | 50.2 | 50.5 | 51.3 | 50.1 | 50.5 | **51.7** | **51.3** | 50.6 |
| Min-K%++ | **50.4** | 50.2 | 50.0 | 50.7 | 50.1 | 50.2 | 50.8 | 51.0 | 50.3 |
| ReCALL | 50.2 | 50.9 | **51.0** | 50.7 | 50.1 | 50.4 | 51.0 | 51.0 | 50.6 |
| ReCALL (Hinge) | 50.3 | **51.4** | 50.2 | **51.9** | **50.2** | **50.7** | 50.2 | 51.0 | **50.7** |
| Hinge | 50.1 | 50.2 | 50.2 | 50.9 | 50.1 | 50.0 | 50.7 | 51.0 | 50.3 |

# F   FURTHER THEORY AND PROOFS

First, we state a useful definition and Lemma by Steinke et al. (2023), and then use them to prove Theorem 1.

**Definition 1** (Stochastic Dominance). *[Definition 4.8, Steinke et al. (2023)] Let $X, Y \in \mathbb{R}$ be random variables. We say $X$ is stochastically dominated by $Y$ if $\mathbb{P}[X > t] \leq \mathbb{P}[Y > t]$ for all $t \in \mathbb{R}$.*

**Lemma 1.** *[Lemma 4.9, Steinke et al. (2023)] Suppose $X_1$ is stochastically dominated by $Y_1$. Suppose that, for all $x \in \mathbb{R}$, the conditional distribution $X_2 | X_1 = x$ is stochastically dominated by $Y_2$. Assume that $Y_1$ and $Y_2$ are independent. Then, $X_1 + X_2$ is stochastically dominated by $Y_1 + Y_2$.*

Here, we have the proof of Theorem 1.

*Proof.* Our analysis is similar to Proposition 5.1 by Steinke et al. (2023).
Fix some $t \in \Sigma_1 \times \cdots \times \Sigma_m$, and $i \in [m]$, $a \in V_i$, and $s_{<i} \in V_1 \times \cdots \times V_i$. Using Bayes' law and

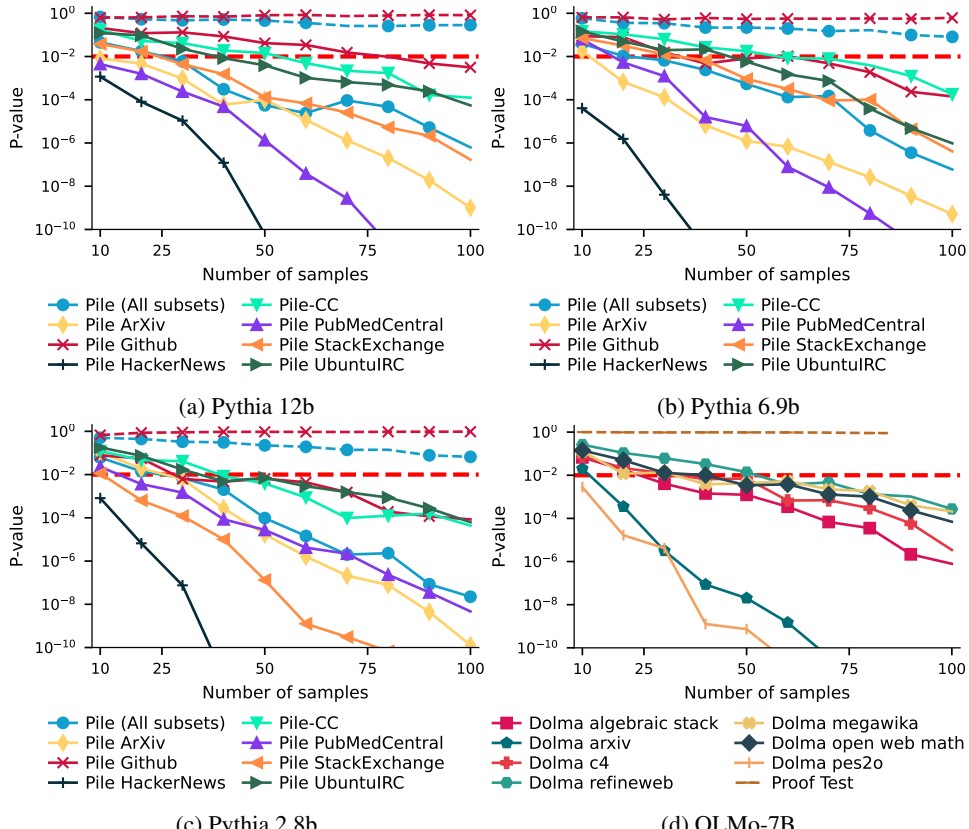

(a) Pythia 12b  (b) Pythia 6.9b

(c) Pythia 2.8b  (d) OLMo-7B

Figure 3: The p-value for different Pythia models and OLMo on subsets of the Pile or Dolma datasets, respectively. We show results for different numbers of samples in the suspect set. For the Pythia models, the solid lines show the training subsets, while the dashed lines are for test subsets (not included in training). The Proof Pile 2 Test subset has less than 100 NIDs. Hence, their lines are plotted only until the highest number of samples is available. We observe that for training sets, the p-value is overall decreasing with the number of samples, enabling the detection of the private data in the model's training set. The test set's p-values are constant, suggesting that no false positives are achieved.

Table 4: **NID-benchmark for Pythia-6.9b.** The AUC for MIAs between the NIDs and the corresponding GIDs on various subsets of the Pile dataset.

| MIA | Pile | | Github | | StackExchange | | UbuntuIRC | Wikipediaen | Train | | | | | Average | |
| | Train | Test | Train | Test | Train | Test | | | PubMedCentral | HackerNews | Pile-CC | ArXiv | | Train | Test |
|---|---|---|---|---|---|---|---|---|---|---|---|---|---|---|---|
| Loss | 57.6 | 50.4 | **69.9** | 51.1 | 50.3 | 50.6 | 50.3 | 50.7 | 50.7 | 61.7 | 50.8 | 50.6 | | 54.7 | 50.7 |
| Min-K% | 56.0 | 51.0 | 65.7 | 50.5 | 50.8 | **51.4** | 50.9 | 50.6 | 50.9 | 63.2 | 51.8 | 50.7 | | 54.5 | 51.0 |
| Min-K%++ | 55.1 | 51.3 | 69.3 | 50.5 | **51.3** | 50.4 | **51.4** | 51.8 | 51.6 | 74.5 | **52.8** | 51.8 | | 56.6 | 50.7 |
| ReCALL | 52.4 | **51.4** | 55.9 | 51.1 | 50.1 | 51.0 | 50.1 | 50.5 | 50.4 | 60.3 | 50.3 | 50.7 | | 52.3 | 51.2 |
| ReCALL (Hinge) | 51.2 | 50.6 | 53.2 | 51.2 | 50.1 | 50.1 | 51.0 | 50.9 | 50.1 | 52.6 | 50.0 | 50.0 | | 51.0 | 50.6 |
| Hinge | **57.7** | 50.7 | 69.9 | **51.6** | 50.4 | 50.1 | 50.2 | 50.0 | 50.7 | 61.7 | 50.7 | 50.3 | | 54.6 | 50.8 |

$\varepsilon$-DP, we have

$$\mathbb{P}[S_i = a | M(S) = t, S_{<i} = s_{<i}]$$

$$= \frac{\mathbb{P}[M(S) = t | S_i = a, S_{<i} = s_{<i}] \mathbb{P}[S_i = a]}{\mathbb{P}[M(S) = t | S_{<i} = s_{<i}]}$$

$$= \frac{\mathbb{P}[M(S) = t | S_i = a, S_{<i} = s_{<i}] \frac{1}{|V_i|}}{\sum_{j=1}^{|V_i|} \mathbb{P}[M(S) = t | S_i = V_{i,j}, S_{<i} = s_{<i}] \mathbb{P}[S_i = V_{i,j}]}$$

$$= \frac{\mathbb{P}[M(S) = t | S_i = a, S_{<i} = s_{<i}] \frac{1}{|V_i|}}{\sum_{j=1}^{|V_i|} \mathbb{P}[M(S) = t | S_i = V_{i,j}, S_{<i} = s_{<i}] \frac{1}{|V_i|}}$$

$$= \frac{1}{1 + \sum_{j=1, V_{i,j} \neq a}^{|V_i|} \frac{\mathbb{P}[M(S) = t | S_i = V_{i,j}, S_{<i} = s_{<i}]}{\mathbb{P}[M(S) = t | S_i = a, S_{<i} = s_{<i}]}} \in \left[ \frac{1}{1 + (|V_i| - 1)e^{\varepsilon}}, \frac{e^{\varepsilon}}{|V_i| - 1 + e^{\varepsilon}} \right]$$

Table 5: **NID-benchmark for Pythia-2.8b.** The AUC for MIAs between the NIDs and the corresponding GIDs on various subsets of the Pile dataset.

| MIA | Pile | | Github | | StackExchange | | UbuntuIRC | Wikipediaen | Train | HackerNews | Pile-CC | ArXiv | Average | |
|---|---|---|---|---|---|---|---|---|---|---|---|---|---|---|
| | Train | Test | Train | Test | Train | Test | | | PubMedCentral | | | | Train | Test |
| Loss | 52.8 | 50.0 | 58.9 | 50.4 | 50.2 | 50.2 | 50.1 | 50.5 | 50.5 | 60.3 | 50.8 | 50.6 | 52.8 | 50.2 |
| Min-K% | 52.1 | **52.4** | 59.5 | **52.9** | 50.6 | 50.3 | 50.2 | 50.1 | **50.6** | 61.6 | 51.6 | 50.5 | 53.0 | 51.8 |
| Min-K%++ | 50.3 | 52.3 | 58.2 | 50.6 | **50.9** | 50.1 | 50.2 | 50.2 | 50.3 | **73.6** | **52.8** | **51.4** | 54.2 | 51.0 |
| ReCALL | **53.7** | 51.1 | **64.4** | 52.2 | 50.1 | 50.1 | 50.2 | 50.8 | 50.5 | 58.0 | 50.2 | 51.1 | 53.2 | 51.2 |
| ReCALL (Hinge) | 50.9 | 50.6 | 50.9 | 50.8 | 50.5 | **50.7** | **50.9** | 52.3 | 50.2 | 51.3 | 50.2 | 50.1 | 50.8 | 50.7 |
| Hinge | 53.0 | 50.4 | 58.9 | 51.1 | 50.3 | 50.3 | 50.2 | 50.2 | 50.5 | 59.9 | 50.7 | 50.4 | 52.7 | 50.6 |

Table 6: **NID-benchmark for Pythia-12b.** The TPR @ 1% FPR for MIAs between the NIDs and the corresponding GIDs on various subsets of the Pile dataset.

| MIA | Pile | | Github | | StackExchange | | UbuntuIRC | Wikipediaen | Train | HackerNews | Pile-CC | ArXiv | Average | |
|---|---|---|---|---|---|---|---|---|---|---|---|---|---|---|
| | Train | Test | Train | Test | Train | Test | | | PubMedCentral | | | | Train | Test |
| Loss | 1.2 | 0.0 | 1.9 | 0.0 | 1.0 | 0.1 | 0.0 | 0.1 | 0.5 | 0.1 | 0.9 | 0.3 | 0.7 | 0.0 |
| Min-K% | 1.1 | 0.0 | 1.6 | 0.0 | **1.0** | **1.8** | 0.3 | 0.9 | 1.0 | 0.2 | 0.9 | 0.6 | 0.9 | 0.6 |
| Min-K%++ | **1.3** | 1.1 | **2.0** | 1.1 | 0.8 | 1.3 | 0.4 | 0.9 | 1.9 | 0.8 | 1.3 | 0.4 | 1.1 | 1.2 |
| ReCALL | 1.2 | 0.2 | 1.5 | 0.0 | 1.0 | 1.5 | **1.4** | 0.7 | 0.8 | 0.9 | **1.9** | 1.0 | 1.1 | 0.5 |
| ReCALL (Hinge) | 1.1 | **1.2** | 1.9 | **1.5** | 0.6 | 1.3 | 0.5 | **1.0** | 0.1 | **1.5** | 1.3 | **2.8** | 1.2 | 1.3 |
| Hinge | 0.0 | 0.4 | 0.0 | 0.5 | 0.9 | 1.5 | 0.5 | 0.5 | **2.1** | 1.1 | 0.9 | 1.3 | 0.8 | 0.8 |

Additionally, we can observe that for all $i \in [m]$, we have that $\mathbb{P}[\mathrm{rank}(t_i, S_i) \leq r_i] = \sum_{j=1}^{r_i} \mathbb{P}[S_i = t_{i,j}]$. Therefore, we can bound

$$\mathbb{P}[\mathrm{rank}(t_i, S_i) \leq r_i] = \sum_{j=1}^{r_i} \mathbb{P}[S_i = t_{i,j} | M(S) = t, S_{<i} = s_{<i}]$$

$$\frac{1}{1 + (|V_i| - 1)e^\varepsilon} \leq \mathbb{P}[S_i = t_i, j \mid M(S) = t, S_{<i} = s_{<i}] \leq \cdot \frac{e^\varepsilon}{|V_i| - 1 + e^\varepsilon}$$

$$\frac{r_i}{1 + (|V_i| - 1)e^\varepsilon} \leq \mathbb{P}[\mathrm{rank}(t_i, S_i) \leq r_i \mid M(S) = t, S_{<i} = s_{<i}] \leq \frac{r_i e^\varepsilon}{|V_i| - 1 + e^\varepsilon}$$

$$\mathbb{P}[\mathrm{rank}(t_i, S_i) \leq r_i | M(S) = t, S_{<i} = s_{<i}] \in \left[ \frac{r_i}{1 + (|V_i| - 1)e^\varepsilon}, \frac{r_i e^\varepsilon}{|V_i| - 1 + e^\varepsilon} \right]$$

Thus, $\mathbb{P}[\mathrm{rank}(t_i, S_i) \leq r_i | M(S) = t, S_{<i} = s_{<i}] \leq \frac{r_i e^\varepsilon}{|V_i| - 1 + e^\varepsilon} = \frac{r_i e^\varepsilon}{e^\varepsilon + |V_i| - 1}$. With that, we can prove the result by induction. We inductively assume that $W_{m-1} := \sum_{i=1}^{m-1} \mathbb{1}[\mathrm{rank}(t_i, S_i) \leq r_i]$ is stochastically dominated by $\hat{W}$ which is Bernoulli$(\frac{r_i e^\varepsilon}{|V_i| - 1 + e^\varepsilon})^{m-1}$. As above, $\mathbb{1}[\mathrm{rank}(t_i, S_i) \leq r_i]$ is statistically dominated by Bernoulli$(\frac{r_m e^\varepsilon}{e^\varepsilon + |V_m| - 1})$. By Lemma 4.9 by Steinke et al. (2023), $W_m = W_{m-1} + \mathbb{1}[\mathrm{rank}(t_m, S_m) \leq r_m]$ is statistically dominated by Bernoulli$(\frac{r_i e^\varepsilon}{|V_i| - 1 + e^\varepsilon})_{i=1}^{m}$.

$\square$

To show the case $(\varepsilon, \delta)$-DP, we will first state Lemma 5.6 by Steinke et al. (2023). Then following the analysis of Proposition 5.7 and Theorem 5.2 by Steinke et al. (2023), we prove Theorem 2.

**Lemma 2.** *[Lemma 5.6, Steinke et al. (2023)] Let $P$ and $Q$ be probability distributions over $\mathcal{Y}$. Fix $\epsilon, \delta \geq 0$. Suppose that, for all measurable $S \subseteq \mathcal{Y}$, we have*

$$P(S) \leq e^\epsilon \cdot Q(S) + \delta \quad and \quad Q(S) \leq e^\epsilon \cdot P(S) + \delta.$$

*Then there exists a randomized function $E_{P,Q} : \mathcal{Y} \to \{0, 1\}$ with the following properties.*

*Fix $p \in [0, 1]$ and suppose $X \sim$ Bernoulli$(p)$. If $X = 1$, sample $Y \sim P$; and, if $X = 0$, sample $Y \sim Q$. Then, for all $y \in \mathcal{Y}$, we have*

$$\mathbb{P}_{X \sim \mathrm{Bernoulli}(p), Y \sim XP + (1-X)Q}\big[X = 1 \wedge E_{P,Q}(Y) = 1 \mid Y = y\big] \leq \frac{p}{p + (1 - p)e^{-\epsilon}}.$$

*Furthermore,*

$$\mathbb{E}_{Y \sim P}[E_{P,Q}(Y)] \geq 1 - \delta \quad and \quad \mathbb{E}_{Y \sim Q}[E_{P,Q}(Y)] \leq \delta.$$

Table 7: **NID-benchmark for Pythia-6.9b.** The TPR @ 1% FPR for MIAs between the NIDs and the corresponding GIDs on various subsets of the Pile dataset.

| | Pile | | Github | | StackExchange | | | | Train | | | | ArXiv | Average | |
|---|---|---|---|---|---|---|---|---|---|---|---|---|---|---|---|
| MIA | Train | Test | Train | Test | Train | Test | UbuntuIRC | Wikipediaen | PubMedCentral | HackerNews | Pile-CC | | | Train | Test |
| Loss | 1.2 | 0.1 | 1.9 | 0.0 | 1.0 | 1.3 | 0.4 | 0.0 | 0.3 | 0.3 | 0.5 | 1.0 | | 0.7 | 0.5 |
| Min-K% | 1.1 | 0.1 | 1.6 | 0.0 | **1.3** | 0.7 | 0.5 | 1.0 | 0.9 | 0.3 | 1.0 | 1.3 | | 1.0 | 0.3 |
| Min-K%++ | 0.9 | 0.7 | 1.0 | 0.6 | 1.2 | 1.4 | 0.4 | 1.3 | 0.9 | 0.9 | 0.4 | 1.1 | | 0.9 | 0.9 |
| ReCALL | 1.0 | 0.2 | 1.5 | 0.0 | 1.2 | 1.3 | **1.1** | 0.6 | 1.2 | 1.2 | **1.2** | **2.1** | | 1.2 | 0.5 |
| ReCALL (Hinge) | **1.3** | **1.4** | **2.0** | **1.5** | 0.5 | **2.6** | 0.6 | **2.3** | **1.8** | **3.3** | 1.2 | 1.8 | | 1.6 | 1.9 |
| Hinge | 0.0 | 0.3 | 0.0 | 0.5 | 0.8 | 1.2 | 0.7 | 0.3 | 1.2 | 1.0 | 0.7 | 0.9 | | 0.6 | 0.7 |

Table 8: **NID-benchmark for Pythia-2.8b.** The TPR @ 1% FPR for MIAs between the NIDs and the corresponding GIDs on various subsets of the Pile dataset.

| | Pile | | Github | | StackExchange | | | | Train | | | | ArXiv | Average | |
|---|---|---|---|---|---|---|---|---|---|---|---|---|---|---|---|
| MIA | Train | Test | Train | Test | Train | Test | UbuntuIRC | Wikipediaen | PubMedCentral | HackerNews | Pile-CC | | | Train | Test |
| Loss | 1.1 | 0.0 | 1.4 | 0.0 | 0.9 | 1.3 | 0.4 | 0.0 | 0.8 | 0.1 | 0.6 | 1.0 | | 0.7 | 0.4 |
| Min-K% | 1.1 | 0.0 | 1.2 | 0.0 | **1.1** | 1.4 | 0.4 | **1.1** | 1.1 | 0.3 | 0.7 | 0.7 | | 0.8 | 0.5 |
| Min-K%++ | 0.9 | 0.6 | 1.3 | 0.5 | 0.8 | 1.5 | 0.3 | 1.0 | **2.3** | 0.8 | 1.0 | 0.3 | | 1.0 | 0.9 |
| ReCALL | 0.1 | 0.0 | 0.5 | 0.0 | 1.0 | 0.1 | 1.5 | 0.1 | 0.9 | 0.7 | **1.0** | **1.7** | | 0.8 | 0.0 |
| ReCALL (Hinge) | **1.3** | **0.7** | **1.6** | **1.0** | 0.7 | 0.1 | **1.7** | 0.4 | 0.1 | **2.4** | 0.8 | 1.1 | | 1.1 | 0.6 |
| Hinge | 0.1 | 0.4 | 0.1 | 0.4 | 0.8 | **1.5** | 0.4 | 0.2 | 1.5 | 1.2 | 0.9 | 0.9 | | 0.7 | 0.8 |

**Theorem 2.** *Let $M : V_1 \times ... \times V_m \to \Sigma_1 \times ... \times \Sigma_m$ be an $(\varepsilon, \delta)$-DP mechanism under replacement. Let $S \in V_1 \times ... \times V_m$ be uniformly random. Let $T = M(S) \in \Sigma_1 \times ... \times \Sigma_m$. Then, for all $v \in \mathbb{R}$, all $t \in \Sigma_1 \times ... \times \Sigma_m$ in the support of $T$, and all $r_1, ..., r_m$ with $r_i \leq |V_i|$,*

$$\mathbf{P}_{S \leftarrow V_1 \times ... \times V_m, T = M(S)}[\sum_{i=1}^{m} \mathbb{1}[\mathrm{rank}(t_i, S_i) \leq r_i] \geq v | T = t]$$

$$\leq \beta + \alpha\delta \sum_{i=1}^{m} |V_i|$$

*where*

$$\beta = \mathbf{P}_{\hat{S}}[\hat{S} \geq v],$$

$$\alpha = \max\left(\frac{1}{i}\mathbf{P}_{\hat{S}}[\hat{S} \geq v - i] : i \in \{1, ..., m\}\right),$$

$$\hat{S} \leftarrow \mathrm{Bernoulli}\left(\frac{r_i e^\varepsilon}{|V_i| - 1 + e^\varepsilon}\right)_{i=1}^{m}.$$

Theorem 2 shows the analogous result of Theorem 1 using $(\varepsilon, \delta)$-DP.

Now, we show the proof of Theorem 2.

*Proof.* Our analysis follows Proposition 5.7 and Theorem 5.2 by Steinke et al. (2023).

For $i \in \{0, ..., m\}$ and $s_{\leq i} \in V_1 \times \cdots \times V_i$, let $M(s_{\leq i})$ denote the distribution on $\Sigma_1 \times \cdots \times \Sigma_m$ obtained by conditioning $\bar{M}(S)$ on $S_{\leq i} = s_{\leq i}$. We can express this as a convex combination:

$$M(s_{\leq i}) = \sum_{s_{>i} \in V_i \times \cdots \times V_m} M(s_{\leq i}, s_{>i}) \cdot \mathbb{P}_{S_{>i} \leftarrow V_i \times \cdots \times V_m}[S_{>i} = s_{>i}].$$

Additionally, for all $i \in [m]$, and $a \in V_i$, we define $\hat{M}(s_{\leq i}, a)$ as the distribution on $\Sigma_1 \times \cdots \times \Sigma_m$ obtained by conditioning on $S_{\leq i} = s_{\leq i}$ and $S_{i+1} \neq a$, as follows:

$$\hat{M}(s_{\leq i}, a) = \sum_{b \in V_i, a \neq b} \frac{1}{|V_i| - 1} M(s_{\leq i}, b).$$

We define $S \leftarrow V_1 \times \cdots \times V_m$ to represent uniform sampling over $V_1 \times \cdots \times V_m$. For all $i \in [m]$, we have that the distributions $P$ and $Q$ on $\Sigma_1, ..., \Sigma_m$, and let $E_{P,Q} : \Sigma_1, ..., \Sigma_m \to \{0, 1\}$ be the

Table 9: **NID-benchmark for OLMo 7B.** The TPR @ 1% FPR for MIAs between the NIDs and the corresponding GIDs on various subsets of the Dolma dataset.

| MIA | C4 | PeS2o | MegaWika | ArXiv | refineweb | algebraic stack | open web math | Proof Pile 2 Test | Average Train |
|---|---|---|---|---|---|---|---|---|---|
| Loss | 0.4 | 0.9 | 0.4 | **1.2** | 0.8 | 0.9 | 0.3 | **0.0** | 0.7 |
| Min-K% | 0.7 | 0.5 | **1.5** | 0.3 | 0.9 | 0.5 | 0.4 | 0.0 | 0.7 |
| Min-K%++ | **1.1** | 0.8 | 0.2 | 0.8 | **2.0** | 0.3 | 0.9 | 0.0 | 0.9 |
| ReCALL | 0.7 | 0.6 | 0.6 | 0.7 | 0.7 | 0.9 | 0.6 | 0.0 | 0.7 |
| ReCALL (Hinge) | 0.7 | 0.3 | 1.1 | 0.2 | 0.2 | **1.1** | **2.2** | 0.0 | 0.8 |
| Hinge | 0.9 | **1.0** | 1.0 | 0.6 | 1.1 | 1.1 | 0.9 | 0.0 | 0.9 |

Table 10: **p-values for DI at 100 samples in the suspect data.** To reject the null hypothesis, we use the threshold of 1% for the p-values. All the outcomes from our method are correct (✔).

| Model | Data | Subset | p-value | DI outcome |
|---|---|---|---|---|
| Pythia 12b | Train | Pile (All subsets) | ≤ 0.0001 | ✔ |
| | | Pile Github | 0.0031 | ✔ |
| | | Pile StackExchange | ≤ 0.0001 | ✔ |
| | | Pile HackerNews | ≤ 0.0001 | ✔ |
| | | Pile-CC | 0.0001 | ✔ |
| | | Pile ArXiv | ≤ 0.0001 | ✔ |
| | | Pile PubMedCentral | ≤ 0.0001 | ✔ |
| | | Pile UbuntuIRC | ≤ 0.0001 | ✔ |
| | Test | Pile (All subsets) | 0.2847 | ✔ |
| | | Pile Test Github | 0.8182 | ✔ |
| Pythia 6.9b | Train | Pile (All subsets) | ≤ 0.0001 | ✔ |
| | | Pile Github | 0.0001 | ✔ |
| | | Pile StackExchange | ≤ 0.0001 | ✔ |
| | | Pile HackerNews | ≤ 0.0001 | ✔ |
| | | Pile-CC | 0.0002 | ✔ |
| | | Pile ArXiv | ≤ 0.0001 | ✔ |
| | | Pile PubMedCentral | ≤ 0.0001 | ✔ |
| | | Pile UbuntuIRC | ≤ 0.0001 | ✔ |
| | Test | Pile (All subsets) | 0.0811 | ✔ |
| | | Pile Test Github | 0.6139 | ✔ |
| Pythia 2.8b | Train | Pile (All subsets) | ≤ 0.0001 | ✔ |
| | | Pile Github | ≤ 0.0001 | ✔ |
| | | Pile StackExchange | ≤ 0.0001 | ✔ |
| | | Pile HackerNews | ≤ 0.0001 | ✔ |
| | | Pile-CC | ≤ 0.0001 | ✔ |
| | | Pile ArXiv | ≤ 0.0001 | ✔ |
| | | Pile PubMedCentral | ≤ 0.0001 | ✔ |
| | | Pile UbuntuIRC | ≤ 0.0001 | ✔ |
| | Test | Pile (All subsets) | 0.0660 | ✔ |
| | | Pile Val Github | 0.9632 | ✔ |
| OLMo 7B | Train | Dolma open web math | ≤ 0.0001 | ✔ |
| | | Dolma PeS2o | ≤ 0.0001 | ✔ |
| | | Dolma refineweb | 0.0003 | ✔ |
| | | Dolma algebraic stack | ≤ 0.0001 | ✔ |
| | | Dolma MegaWika | 0.0002 | ✔ |
| | | Dolma arxiv | ≤ 0.0001 | ✔ |
| | | Dolma c4 | ≤ 0.0001 | ✔ |
| | Test | Proof Pile 2 Test | 0.8961 | ✔ |

randomized function given by Lemma 2 (using $p = \frac{1}{|V_i|}$). Specifically, all $s_{\leq i} \in V_1 \times \cdots \times V_i$, all $t \in \Sigma_1 \times \cdots \times \Sigma_m$, and all $a \in V_i$, we have

$$\mathbb{P}_{S \leftarrow V_1 \times \cdots \times V_m, T \leftarrow M(S), E}[S_i = a \land E_{M(s_{<i}, a), \hat{M}(s_{<i}, a)}(T) = 1 | S_{\leq i} = s_{\leq i}, T = t] \leq \frac{e^\varepsilon}{|V_i| - 1 + e^\varepsilon},$$

$$\mathbb{E}_{S \leftarrow V_1 \times \cdots \times V_m, T \leftarrow M(S), E}[E_{M(s_{<i}, a), \hat{M}(s_{<i}, a)}(T) | S_{\leq i} = (s_{<i}, a)] \geq 1 - \delta.$$

For simplicity, for all $i \in [m]$, we define $E_{M(s_{<i}, V_i)}(y) := \prod_{a \in V_i} E_{M(S_{<i}, a), \hat{M}(S_{<i}, a)}(y)$

and, for $b \in V_i$, we have

$$\mathbb{E}_{S \leftarrow V_1 \times \cdots \times V_m, T \leftarrow M(S), E}[E_{M(s_{<i}, V_i)}(T)|S_{\leq i} = (s_{<i}, b)] \geq 1 - |V_i|\delta.$$

For all $a \in V_i$, we define a $j := \text{rank}(t_i, a)$, so we can rewrite

$$\mathbb{P}_{S \leftarrow V_1 \times \cdots \times V_m, T \leftarrow M(S), E}[S_i = a \wedge E_{M(s_{<i}, V_i)}(T) = 1|S_{\leq i} = s_{\leq i}, T = t]$$
$$= \mathbb{P}_{S \leftarrow V_1 \times \cdots \times V_m, T \leftarrow M(S), E}[\text{rank}(t_i, S_i) = j \wedge E_{M(s_{<i}, V_i)}(T) = 1|S_{\leq i} = s_{\leq i}, T = t].$$

Note that there is a bijective relationship between $a$ and $j$. Therefore, we have that

$$\mathbb{P}_{S \leftarrow V_1 \times \cdots \times V_m, T \leftarrow M(S), E}[\text{rank}(t_i, S_i) \leq r_i \wedge E_{M(s_{<i}, V_i)}(T) = 1|S_{\leq i} = s_{\leq i}, T = t] \leq \frac{r_i e^\varepsilon}{|V_i| - 1 + e^\varepsilon}.$$

For $j \in [m]$, $s \in V_i \times \cdots \times V_m$, and $t \in \Sigma_1 \times \cdots \times \Sigma_m$, define

$$\widetilde{W}_j(s, t) := \sum_{i < j} \mathbb{1}[\text{rank}(t_i, S_i) \leq r_i] \cdot E_{M(s_{<i}, V_i)}(t) = \sum_{i < j} \mathbb{1}[\text{rank}(t_i, S_i) \leq r_i \wedge E_{M(s_{<i}, V_i)}(t) = 1]$$

$$\hat{W}_j(t) = \sum_{i \in [j]} S_i(t),$$

where, for each $i \in [m]$ independently, $S(t)_i \leftarrow \text{Bernoulli}\left(\frac{r_i e^\varepsilon}{|V_i| - 1 + e^\varepsilon}\right)$

By induction and Lemma 1, for any $j \in [m]$ and $t \in \Sigma_1 \times \cdots \times \Sigma_m$, the conditional distribution $(\widetilde{W}_m(S, t)|M(S) = t)$ where $S \leftarrow V_1 \times \cdots \times V_m$ is stochastically dominated by $\hat{W}_m(t)$.

For $s \in V_1 \times \cdots \times V_m$ and $t \in \Sigma_1 \times \cdots \times \Sigma_m$, define

$$F(s, t) := \sum_{i=1}^m \mathbb{1}\left[E_{M(s_{<i}, V_i)}(t) = 0\right],$$

so that

$$W_m(s, t) := \sum_{i=1}^m \mathbb{1}[\text{rank}(t_i, S_i) \leq r_i] \leq \hat{W}_m(s, t) + F(s, t).$$

Since the conditional distribution $(W_m(S, t)|M(S) = t)$, where $S \leftarrow V_1 \times \cdots \times V_m$ is stochastically dominated by $W_m(t)$, $W_m$ is stochastically dominated by the convolution $\hat{W}_m(T) + F(S, T)$. Finally, $F(s, t)$ is supported on $\{0, 1, \ldots, m\}$ and

$$\mathbb{E}[F(s, t)] = \sum_{i=1}^m \mathbb{P}[E_{M(s_{<i}, a), \hat{M}(s_{<i}, a)}(T) = 0] \leq \delta \sum_{i=1}^m |V_i|.$$

Since $\hat{W}_m(T)$ does not depend on $S$, the input $S$ does not contribute to the dependence between $F(S, T)$ and $W_m(T)$, so we can elide this input in the statement, that is, $F(T) = F(S, T)$ for $S$ drawn from an appropriate distribution.

Given these constraints, we can formulate finding the optimal distribution $F(t)$ for a given $t \in \Sigma_1 \times \cdots \times \Sigma_m$ and $v \in \mathbb{R}$ as a linear program:

$$\text{maximize} \quad \mathbb{P}_{\check{W}, F}[\check{W}(t) + F(t) \geq v] - \sum_{i=0}^m \mathbb{P}[F(t) = i] \cdot \mathbb{P}[\check{W}(t) \geq v - i]$$

$$\text{subject to} \quad \mathbb{E}_F[F(t)] = \sum_{i=0}^m \mathbb{P}_F[F(t) = i] \cdot i \leq \delta \sum_{i=1}^m |V_i|,$$

$$\sum_{i=0}^m \mathbb{P}_F[F(t) = i] = 1, \text{ and}$$

$$\mathbb{P}_F[F(t) = i] \geq 0 \quad \forall i \in \{0, 1, \ldots, m\},$$

where $\check{W}(t) := \sum_{i=1}^{m} \mathbb{1}[\text{rank}(t_i, S_i) \leq r_i]$ for $S_i \leftarrow \text{Bernoulli}\left(\frac{r_i e^\varepsilon}{|V_i| - 1 + e^\varepsilon}\right)^m$.

By strong duality, the linear program above has the same value as its dual:

$$
\begin{aligned}
&\text{minimize} && \alpha \cdot \delta \sum_{i=1}^{m} |V_i| + \beta \\
&\text{subject to} && \alpha \cdot i + \beta \geq \mathbb{P}_{\check{W}}[\check{W}(t) \geq v - i] \quad \forall i \in \{0, 1, \ldots, m\}, \\
& && \alpha \geq 0.
\end{aligned}
$$

Any feasible solution to the dual gives an upper bound on the primal. So, in particular, we can use the solution provided by

$$
\begin{aligned}
\beta &= \mathbb{P}_{\check{W}^*}[\check{W}^* \geq v], \\
\alpha &= \max\left(\{0\} \cup \left\{\frac{1}{i}\left(\mathbb{P}_{\check{W}^*}[\check{W}^* \geq v - i] - \beta\right) : i \in \{1, 2, \ldots, m\}\right\}\right),
\end{aligned}
$$

where $\check{W}^*$ is a distribution on $\mathbb{R}$ that satisfies $\mathbb{P}_{\check{W}^*}[\check{W}^* \geq v - i] \geq \mathbb{P}_{\check{W}}[\check{W}(t) \geq v - i]$ for all $i \in \{0, 1, \ldots, m\}$ and all $t$ in the support of $T$. $\qquad \square$

## G  OUR PRIVACY AUDITING AT THE EXAMPLE OF RANDOMIZED RESPONSE

To demonstrate the effectiveness of our adapted procedures, we consider the randomized response mechanism (Warner, 1965). Formally, we are given $m$ samples, and each of them corresponds to a private integer value $s_i$ between 1 and $c$, which means $V_i = [c]$, and we use the randomized response mechanism to release these private integers, as

$$
y_i = \begin{cases} s_i & \text{with probability } \frac{1}{c} + \gamma, \\ a & \text{with probability } \frac{1}{c} - \frac{\gamma}{c-1} \end{cases} \quad \forall a \in [c], a \neq s_i.
$$

By choosing $\gamma = \frac{e^\varepsilon - 1}{c(1 + \frac{e^\varepsilon}{c-1})}$, we have a $\varepsilon$-DP mechanism. The auditor ranks the possible $c$ values from the most to the least likely. In this case, the only information that the auditor has to predict $s_i$ is the corresponding output $y_i$. We can observe that $y_i$ is the most likely input $s_i$, therefore the best that the auditor can do is to rank $y_i$ as the most likely value for $s_i$, and the others in random orders, as it does not have any information regarding the other possibilities. We can also observe that it gives the correct answers with probability $\frac{e^\varepsilon}{c-1+e^\varepsilon}$, which is equivalent to the bound obtained from Theorem 1. Figure 4 shows how our method performs for different choices of top-$r$ ranks, and cardinality of the sets $c = |V_i|$. A high cardinality is particularly useful when auditing higher privacy budgets, while for smaller privacy budgets, it increases the required number of samples to obtain a tight privacy result. Therefore, the best cardinality for a given setting depends on the privacy budget, and on the hardness of the task. Furthermore, we highlight that our result is tight for $r_i = 1$ with enough samples.

## H  DP-SGD AUDITING

In the following subsection, we show additional experiments for DP-SGD auditing, and the pseudocode of the auditing procedure.

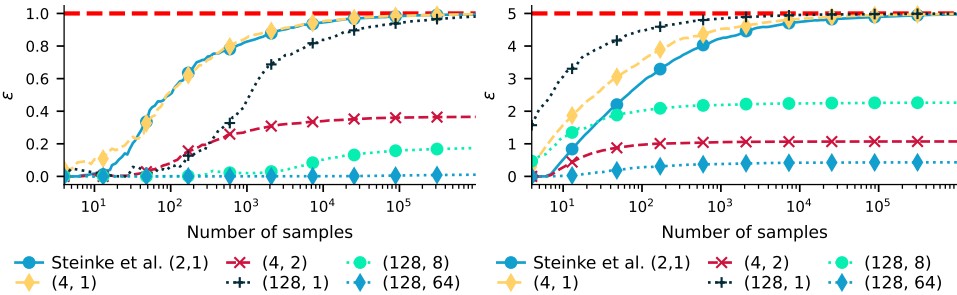

Figure 4: **Randomized response** mechanism with $\varepsilon = \{1, 5\}$. The red dashed line indicates the real $\varepsilon$ of the mechanism, while other ones indicate the estimated lower bound of $\varepsilon$ with 99% confidence for different choices of cardinality $c$, and rank threshold $r$. The (2,1) case corresponds to the method proposed by Steinke et al. (2023). Each label is written as (cardinality $c$, rank threshold $r$).

## H.1 Further Experiments on DP-SGD Auditing

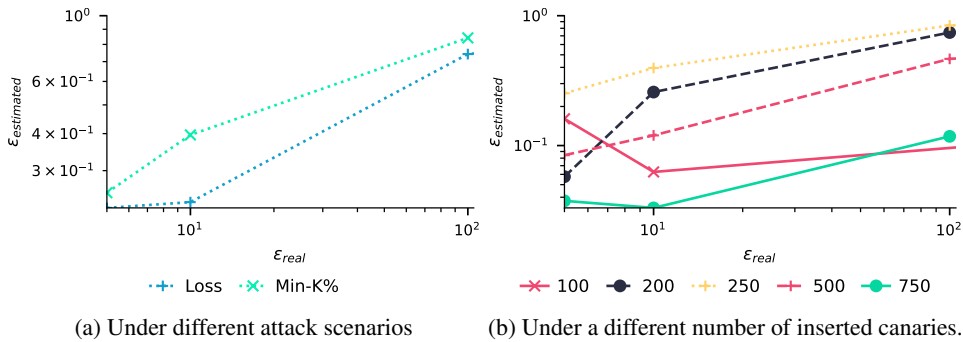

(a) Under different attack scenarios    (b) Under a different number of inserted canaries.

Figure 5: Comparison of impact on $\varepsilon$ estimation versus actual $\varepsilon$

Figure 5a reveals compares various scoring functions used for auditing. The gap between different attacks remains substantial. As shown in Figure 5a, we can observe that $\varepsilon$ estimation improves with increasing number of inserted canaries until 250. With 500 and 750 canaries, the audit becomes looser. These observations align with those of Steinke et al. (2023), who also observed a similar trend for black-box audits. We can agree that one possible explanation may be a black-box analysis gives a weaker theoretical privacy analysis. The other explanation may be utilizing a small model.

## H.2 Pseudocode for DP-SGD auditing

Algorithm 1 summarizes our approach for auditing DP-SGD using the results given by Theorem 2. We highlight that when for all $i \in [m]$, we have $|V_i| = 2$ and $r_i = 1$, the algorithm is equivalent to the fixed-length dataset case proposed by Steinke et al. (2023).

---

**Algorithm 1** Adapted version of the black-box DP-SGD Auditor algorithm proposed by Steinke et al. (2023) for fixed-length dataset with NIDs.

---

**Require:** Dataset $D_0$, sets of canaries $V = \{V_1, \ldots, V_m\}$, the target ranks $r_1, \ldots, r_m$, and the DP-SGD settings
1: **for** $i \in [m]$ **do**
2:     $S_i \leftarrow \text{Unif}\{V_i\}$
3: **end for**
4: $D_1 := \{V_{i,S_i} : i \in [m]\}$
5: $D = D_0 \cup D_1$
6: Run DP-SGD on $D$ with given parameters, yielding $\{w^0, w^1, \ldots, w^\ell\}$
7: **for** $i \in [m]$ **do**
8:     $Y_{i,j} \leftarrow \text{SCORE}(V_{i,j}; w^\ell) \quad \forall j \in [|V_i|]$
9:     $T_i \leftarrow \text{argsort}(Y_{i,j} \forall j \in [|V_i|])$
10: **end for**
11: $c \leftarrow 0$
12: **for** $i \in [m]$ **do**
13:     **if** $T_{i,S_i} \leq r_i$ **then**
14:         $c \leftarrow c + 1$
15:     **end if**
16: **end for**
17: **return** Compute $\varepsilon_{\text{lower}}$ using the formula given by Theorem 2

---

