# OpenReview forum: "Privacy Auditing for Large Language Models with Natural Identifiers"
_ICLR.cc/2025/Workshop/BuildingTrust — Submitted to BuildingTrust_

### Official Review · Reviewer_KSE7 · 2025-03-01
**Paper Review**

**Rating:** 5
**Confidence:** 4

**Review:**

> Summary:

This paper introduces Natural Identifiers as an approach to privacy auditing in large language models, enabling distribution-shift-free membership inference attacks, dataset inference without private validation sets, and post-hoc differential privacy auditing without retraining.

> Pros:

1.The way of conducting LLM privacy audits is new, using new types of identifiers, which can to some extent address certain challenges in LLM privacy regime.


> Cons:

1.Some claims are not well-supported by the experiments. For examples, from line 313 - line 317, how can current results support the claim that models with larger training corpus memorize less?

2.In crafted examples, are the canaries crafted by only replacing original sample NIDs in sentences with new generated NIDs while keeping the rest part unchanged? I think presentation of methodology settings can be polished to be more clear (an overall framework, or figs of examples).

3.The organization of Table 1 is confusing. Why does Table 1 have some first-row entries as dataset names while others are labeled as "Train" and "Average"? Additionally, in the second row, while "Train" and "Test" splits are indicated, multiple dataset names are listed, and the test results are not provided.

4.From my perspective, aside from the introduction of the new benchmark framework itself, the paper does not present surprising results or general takeaways that are different from previous literatures.

---

### Official Review · Reviewer_QBWy · 2025-03-02
**Review of Proposed Privacy Auditing Approach for LLMs**

**Rating:** 5
**Confidence:** 3

**Review:**

# Summary

The paper suggests "Natural Identifiers" (NIDs) as an alternative solution to address privacy auditing challenges in large language models (LLMs). The authors demonstrate that existing methods of privacy auditing like membership inference attacks (MIAs) and dataset inference have limitations in that they need non-member data to be identically distributed as training data and are reliant on canary data planted during training, thereby posing challenges in auditing pre-trained LLMs.

NIDs, as proposed, are randomly generated strings (the authors give some examples like SSH keys, crypto hashes) which should appear by chance in LLM training datasets. The observation is that these NIDs allow for constructing "unbounded additional random strings from the same distribution, which the authors contend can be used as non-members or alternative canaries for audit. This enables robust testing of MIAs, dataset inference on any suspect set with NIDs, and post-hoc privacy auditing without retraining.".

The paper demonstrates the effectiveness of NIDs by measuring MIAs and inferring from datasets for Pythia family of models and Pile dataset and OLMo models. The result is that NIDs facilitate privacy auditing and analysis of privacy risks in LLMs without requiring retraining and generating accurate results.

# Quality and Clarity of the Paper

Generally, the paper is well-written and organized. Overall quality seems to be good.


# Pros and Cons
## Pros

1. **Fair Novelty:** The paper introduces a fairly novel concept; utilizing natural identifiers within training sets as both a baseline and as an auditing mechanism post-hoc sounds like an acceptable contribution because the prior work has relied upon injecting synthetic canaries or train/test splits.
2. **Practical Applicability:** The proposed approach, tackling the central problem of lacking suitable non-member data for proper MIA testing, is practically applicable in the sense that it avoids expensive retraining of LLMs and can be applied directly to current pre-trained models.

## Cons:
1. **Distribution Assumption for NIDs:** NIDs in real-world applications are not necessarily random strings because they are part of a context. In the case of hashes, a source hash could be the hash of a trending file, a specific version of software, or even an indexed sensitive document on the internet. Because of this, it could be presented to the LLMs in many textual scenarios during training. Replacing it with an actually random hash shatters these associations and semantic relationships learned. The LLM may have witnessed the original hash in diverse text contexts (security alerts, codebases, threads), and such contexts enrich its understanding of the information being transmitted by that NID. A random substitution lacks this rich context information, yielding a distribution difference that the attacker may be able to exploit or renders the auditing unreliable.
2. **Scalability:** Identifying NIDs in massive datasets is computationally expensive. While the paper claims this is easier than retraining, it might still be a significant bottleneck, particularly for real-time monitoring. How does the identification process scale? Are there efficient algorithms for identifying new or less common NID types?

3. **Limited Range of Identifiers:** The approach relies on the availability of specific, defined NID forms. Most sensitive data elements in LLM training data are likely not structured identifiers such as names, addresses, and opinions, for which the proposed approach is not found effective.

# Minor Errors:
- Dictation error for "divers" on line 54.

# Questions
1. Why can NIDs have meanings in the context in which they occur but truly random strings cannot? (line 194)

---

### Decision · Program_Chairs · 2025-03-04

**Decision:**

Reject

**Comment:**

The limitations of standard differential privacy auditing techniques are well pointed out by the authors and furthermore, the novelty of the proposed solution is quite significant. However, the reviewers have raised some good questions for the alternative solution to be viable. In particular, identification of NID's in a massive dataset and their limited range are valid points.